# The Joint Application of Phosphorus and Ammonium Enhances Soybean Root Growth and P Uptake

**Ciro Antonio Rosolem** *[ID], **Thiago Barbosa Batista, Patrícia Pereira Dias, Laudelino Vieira da Motta Neto** [ID] **and Juliano Carlos Calonego**

Department of Crop Science, School of Agricultural Sciences, São Paulo State University, 3780 Universitaria Av., Botucatu 18610-034, Brazil; batista.thiagob@gmail.com (T.B.B.); patricia.dias@unesp.br (P.P.D.); l.mota@unesp.br (L.V.d.M.N.); juliano.calonego@unesp.br (J.C.C.)
* Correspondence: ciro.rosolem@unesp.br; Tel.: +55-14-9775-1083

**Abstract:** It has been shown that the joint application of phosphorus (P) and ammonium (N-NH$_4^+$) increases maize root proliferation and P acquisition by maize in alkaline soils, but this has not been shown in acidic soils for legumes. A greenhouse experiment was conducted to assess the effect of the joint application of P and NH$_4^+$ on soybean root growth and P acquisition. Soybean was grown in glass-walled pots without P, with monoammonium phosphate (MAP) and triple super phosphate (TSP) applied on the soil surface or localized. The soil P increased irrespective of the P source and localization. The rhizosphere pH was decreased by MAP, while the soil bulk pH was not affected. The TSP increased the root length by 55% and MAP by 76% over the control, and the number of root tips increased by 21% with TSP, 58% with MAP applied on the soil surface, and 78% with MAP localized. The soybean dry matter, N and P uptake, and P use efficiency were increased by P fertilization, mainly with MAP localized. The joint application of P and ammonium decreases the soybean rhizosphere pH, which results in root proliferation early in the cycle, and eventually in higher P uptake and use efficiency.

**Keywords:** *Glycine max*; root architecture; phosphorus acquisition; phosphate fertilizer

## 1. Introduction

Phosphorus (P) is relatively immobile in tropical weathered soils that are rich in aluminum and iron oxides. The P diffusion in mineral soils varies from $10^{-12}$ to $10^{-15}$ m$^2$ s$^{-1}$ [1], which is considered to be very slow. Thus, the higher the distance from the P source to the roots, the higher the P fixation onto the mineral matrix, which decreases the P availability roots and results in P depletion zones in the rhizosphere [2,3]. Hence, root architecture, mainly root length, is one of the most important factors for soil P acquisition [4,5]. Despite the observation that an increased root growth rate does not always compensate for the low P diffusion in low P soils [6], topsoil foraging by roots is important for P acquisition [7]. Consequently, the proliferation of lateral roots to increase soil exploration is an important plant strategy to enhance the uptake of immobile nutrients such P [8], what has been reported for common beans and soybeans in soil where the P concentration was stratified [9]. It has been shown in rice that genotypes with early root growth and root proliferation had a higher ability to mine soil P [10]. Therefore, it is clear that root proliferation is an important mechanism of P acquisition by plants.

It has been shown [2,11] that the joint application of P and ammonium increased the root proliferation in nutrient-rich patches, and this combination of elements also increased the P uptake as a consequence of the rhizosphere pH reduction; however, that response was not reported where N was supplied as nitrate or urea. As root architecture is deemed to be an important strategy for P acquisition, understanding the relationship between these elements is important to support the decision-making of farmers to increase P-use efficiency and soybean yields in cropped areas.

Ammonium has been shown to shape plant root architecture by triggering lateral root emergence, which results in a highly branched root system [12]. The acidification of the root apoplast as ammonium is absorbed by the plant increases protonated auxin import into cortical cells and stimulates root branching [13]. The hypothesis has been raised [2] that root proliferation relied on an ammonium-dependent mechanism, which would be responsible for the rhizosphere acidification that led to root branching and higher P uptake. This was proved to be the case in maize, both under alkaline and acidic soils [3,11,12]. Ma and coauthors [3] reported a positive interaction on nutrient uptake and root proliferation in alkaline nutrient-rich soil patches where ammonium and P were supplied locally. Though the effect of ammonium has been demonstrated in maize, it is unclear if a legume would show a similar response.

Root proliferation has been shown to be an important strategy to acquire soil P in common beans and soybeans, since genotypes with higher root branching in the uppermost soil layer were more efficient [9]. Therefore, the question is, would the joint application of ammonium and P enhance root growth in soybean, and would this influence plant growth? It has been demonstrated that N application to soybean hinders the biological nitrogen fixation [14], but it is also known that N fertilization has an inconsistent and small effect on soybean seed yield, depending on the rate, time of application, and management system [15,16]. Therefore, we hypothesized that (i) the application of ammonium and P jointly could change the root architecture, increase P acquisition, and soybean early growth as a result of a lower rhizosphere pH; and (ii) the forms of application could undermine the P-use efficiency (PUE) and plant performance. It could be argued that N fertilization is not usually recommended for soybean [17], but the biological nitrogen fixation is only effective after three weeks [15], and an early establishment of a vigorous root system is paramount, mainly in P fixing tropical soils. The objective was to evaluate P sources applied locally or broadcasted on soybean root and shoot growth, P-use efficiency and rhizosphere pH and unravel a win-win mechanism to the joint supply of P and ammonium to acid tropical soils.

## 2. Materials and Methods

### 2.1. Plant Growth

The experiment was carried out in a greenhouse, using PVC cylinders cut in half longitudinally, 58 cm high and 13 cm radius. Each experimental unit was filled with a clayey Rhodic Hapludox [18], with 614 g $kg^{-1}$ clay, 196 g $kg^{-1}$ silt, and 190 g $kg^{-1}$ sand from a fallowed area. The chemical properties of the soil, collected at a depth up to 0–20 cm, were as follows: pH ($CaCl_2$), 4.7; Presin, 6.0 mg $dm^{-3}$; OM, 25.0 mg $dm^{-3}$; H + $Al^{3+}$, 48.0 mmol $dm^{-3}$; K, 0.70 mmol $dm^{-3}$; $Ca^{2+}$, 16.0 mmol $dm^{-3}$; $Mg^{2+}$, 8.0 mmol $dm^{-3}$; BS, 25.0 mmol $dm^{-3}$; V (%), 35; and CEC, 73.0 mmolc $dm^{-3}$ [19]. Lime was applied to the soil at a rate of 1.92 Mg $ha^{-1}$ to raise soil cation saturation to 60% [20] sixty days before setting up the experiment.

Two soybean seeds of the cultivar Syn 1562 IPRO—maturity group 6.2 and indeterminate growth habit—were sown at the depth of 4 cm. Before sowing, soybean seeds were inoculated with a culture of *Bradyrhizobium elkani* (strains SEMIA587 and SEMIA 5019) at a concentration of 5 $10^9$ Rhizobia per mL, applied to the seeds at a rate of 1.0 mL 0.5 $kg^{-1}$ of seeds [15]. Each seed was set 9 cm from each other. After emergence, the plants were grown for 25 days, with a nighttime average temperature of $23 \pm 5\,^{\circ}C$ and a daytime average temperature of $33 \pm 5\,^{\circ}C$.

### 2.2. Treatments and Experimental Design

The experiment comprised five treatments in complete randomized blocks, with four replicates. The treatments were as follows: (i) control (without P application), (ii) monoammonium phosphate applied locally beside and below the seeds ($MAP_L$), (iii) monoammonium phosphate broadcasted at the soil surface ($MAP_B$), (iv) triple superphosphate applied locally aside and down the seed furrow ($SFT_L$), and (v) triple superphosphate broadcasted at the soil surface ($SFT_B$). The phosphorus (P) rate calculated was 120 kg $ha^{-1}$ of P, supplied

as monoammonium phosphate (MAP) and triple superphosphate (SFT). Potassium (K) rate (80 kg K ha$^{-1}$) was supplied as potassium chloride. Both P (MAP$_L$ and SFT$_L$ treatments) and K sources were applied at a depth of 12 cm, beside the seed furrow. However, in the MAP$_B$ and SFT$_B$ treatments, just K was applied at the furrow, while P was broadcasted on the soil surface.

### 2.3. Shoot and Root Measurements

At harvest, soybean shoots were cut at ground level, and roots were separated from the soil, using tap water over a 0.5 mm screen. The roots were scanned at 300 dpi, and root superficial area, volume, length, and root tip number were analyzed by using the software WinRHIZO TM version 3.8-b (Regent Instrument Inc., Quebec City, QC, Canada). Soybean shoots and roots were packed in paper bags and dried in an air-forced chamber for 72 h at 65 °C. Then both the shoot (SDM) and root (RDM) dry mass were measured.

### 2.4. Soil Properties and Plant N and P

Before dismantling the pots, two individual soil samples were taken at the depth 0–20 cm and mixed into a composite sample, which was packed in paper bags and dried at shadow and fresh air. Soybean leaves were ground in a Wiley mill and passed through a 0.50 mm sieve for P and N measurements, using the blue molybdate–ascorbic acid method and Kjeldahl method, respectively [21]. Bulk soil pH was measured in dry samples (2 mm sieve) at a soil: solution ratio of 1:2.5 after shaking 0.5 g of soil with CaCl$_2$ solution (10 mmol L$^{-1}$) for 1 h [19]. The rhizosphere pH (pH equipment Tec7-Tecnal—0.001 sensitivity) was determined by shaking off excess soil adhering root surface and colleting just soil tightly adhered to the soybean root surface [11]. Soil P content was assessed with the blue molybdate–ascorbic acid method [19].

Phosphorus-use efficiency (PUE) was calculated by using the SDM and the applied amount of P$_2$O$_5$ [22].

### 2.5. Statistical Analysis

After testing for homogeneity and homoscedasticity data were submitted to ANOVA ($p < 0.05$), and when the difference was significant, the means were compared by using an LSD test. Analyses were run in the software Sisvar 5.6 [23].

### 3. Results

There was no change in the bulk soil pH, regardless of the source and form of fertilizer application; however, the rhizosphere pH was decreased by MAP by 0.25 units on average compared with the control and treatments with TSP (Table 1).

**Table 1.** The pH values to a Rhodic Hapludox under different P sources.

| Treatment | pH * | Rhizosphere pH * |
|:---:|:---:|:---:|
| Control | 5.29 a | 5.56 a |
| MAP$_L$ | 5.40 a | 5.30 b |
| MAP$_B$ | 5.44 a | 5.34 b |
| TSP$_L$ | 5.59 a | 5.50 a |
| TSP$_B$ | 5.59 a | 5.59 a |

* Different letters in each column denote significant difference by the LSD test ($p \leq 0.05$). MAP$_L$ = MAP applied locally within the soil; MAP$_B$ = MAP broadcasted at the soil surface; TSP$_L$ = TSP applied locally within the soil; TSP$_B$ = TSP broadcasted at the soil surface.

Soybean root growth responded to P application (Table 2), and the root length and superficial area were higher with MAP compared with TSP. The greatest number of root tips was observed when MAP was localized, followed by MAP broadcasted. On average, MAP increased root tips by 168% over the control and 21% over STP. Even though the

TSP did not reduce the rhizosphere and bulk soil pH, the localized supply increased the total P uptake (Table 3). As expected, the lowest total P content was observed with no P application. Additionally, the P uptake was increased when the fertilizer was localized compared with broadcasted, regardless of the fertilizer source.

**Table 2.** Soybean root growth, superficial area ($cm^2$), root tip number, volume ($cm^3$), and root length (cm) at the end of the cropping period (25 days).

| Treatment | Root Length cm | Volume $cm^3$ | Superficial Area $cm^2$ | Root Tip Number |
|---|---|---|---|---|
| Control | 75.0 c* | 0.12 c | 10.6 c | 89.3 d |
| $MAP_L$ | 132.5 a | 0.21 a | 19.0 a | 248.8 a |
| $MAP_B$ | 132.3 a | 0.18 ab | 16.7 ab | 230.7 b |
| $TSP_L$ | 114.8 b | 0.19 ab | 16.2 b | 198.9 c |
| $TSP_B$ | 118.0 b | 0.17 b | 15.9 b | 196.6 c |

* Different letters in each column denote significant difference by the LSD test ($p \leq 0.05$). $MAP_L$ = MAP applied locally within the soil; $MAP_B$ = MAP broadcasted at the soil surface; $TSP_L$ = TSP applied locally within the soil; $TSP_B$ = TSP broadcasted at the soil surface.

**Table 3.** Total P content determined at the end of the cropping period (25 days).

| Treatment | Total P Content (g kg$^{-1}$) |
|---|---|
| Control | 2.11 c* |
| $MAP_L$ | 4.71 a |
| $MAP_B$ | 3.81 b |
| $TSP_L$ | 4.18 ab |
| $TSP_B$ | 3.67 b |

* Different letters in each column denote significant difference by the LSD test ($p \leq 0.05$). $MAP_L$ = MAP applied locally within the soil; $MAP_B$ = MAP broadcasted at the soil surface; $TSP_L$ = TSP applied locally within the soil; $TSP_B$ = TSP broadcasted at the soil surface.

Soybean dry matter yields were higher in the presence of the P fertilizer (Figure 1), and the response was more evident in the shoots. The highest shoot dry matter yield was observed with MAP localized, followed by MAP broadcasted, TSP localized, and TSP broadcasted.

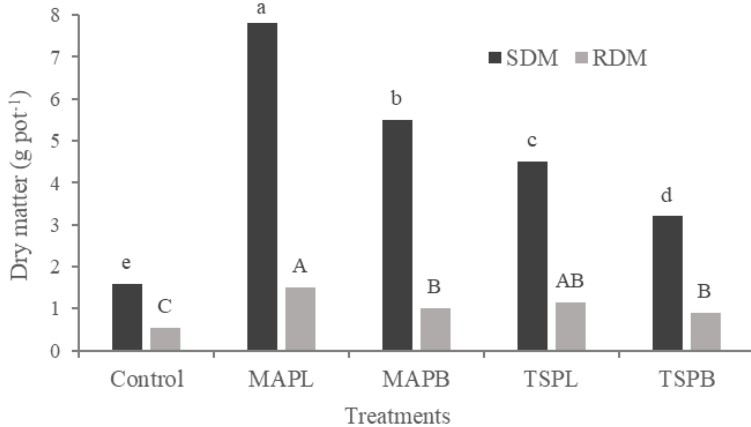

**Figure 1.** Soybean-shoot dry mass (SDM) and root dry mass (RDM) in response to different P sources and application forms. Different letters show significant differences (LSD, $p \leq 0.05$), capitals show differences in shoot dry matter, and lowercase show differences in root dry matter. $MAP_L$ = MAP applied close to the seed furrow; $MAP_B$ = MAP broadcasted at the soil surface; $TSP_L$ = TSP applied close to the seed furrow; $TSP_B$ = TSP broadcasted at the soil surface.

The concentrations of P and N in soybean leaves (Figure 2) were highest where MAP was supplied locally, and the trend was similar to the results obtained for P uptake (Table 3).

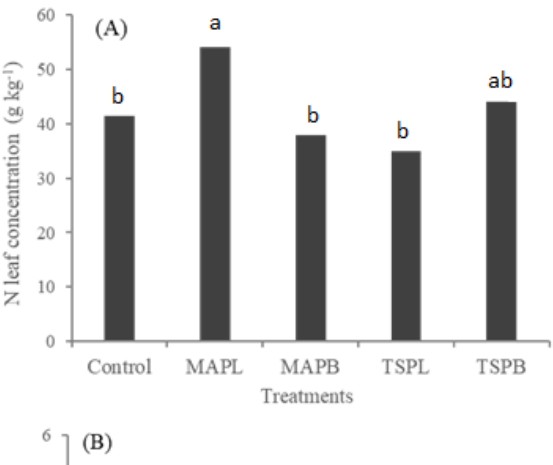

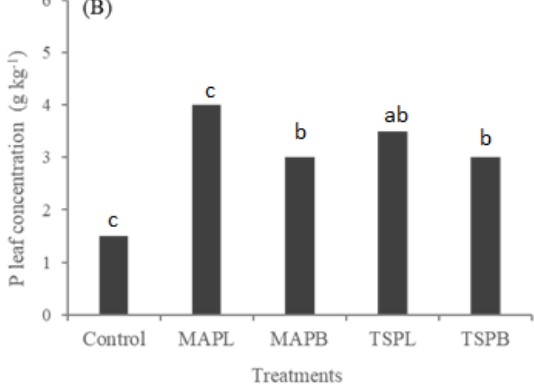

**Figure 2.** Soybean P (**A**) and N leaf content (**B**) as affected by P sources and application forms. Different letters denote significant difference (LSD, $p \leq 0.05$). MAP$_L$ = MAP applied close to the seed furrow; MAP$_B$ = MAP broadcasted at the soil surface; TSP$_L$ = TSP applied locally within the soil; TSP$_B$ = TSP broadcasted at the soil surface.

The greatest PUE was observed with localized MAP, followed by the broadcasted MAP, localized TSP, and TSP broadcasted (Figure 3). As observed for shoot dry matter yields, supplying P with either MAP or SFT locally resulted in higher PUE compared with superficial application.

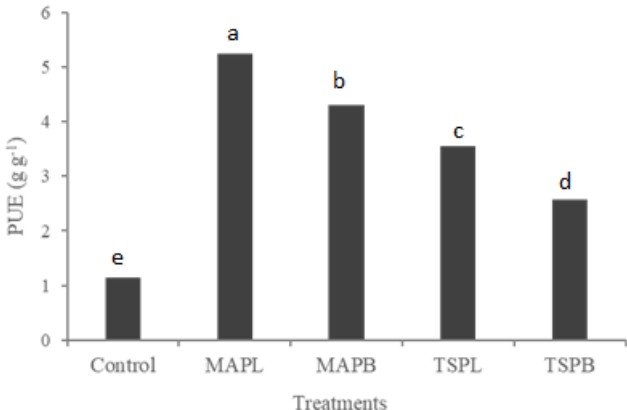

**Figure 3.** Soybean P-use efficiency (PUE) under two P sources and different forms of P supply. Different letters show significant differences (LSD, $p \leq 0.05$). MAP$_L$ = MAP applied locally within the soil; MAP$_B$ = MAP broadcasted at the soil surface; TSP$_L$ = TSP applied locally within the soil; TSP$_B$ = TSP broadcasted at the soil surface.

## 4. Discussion

### 4.1. Root Architecture and Rhizosphere Acidification

The joint effect of nutrients and application strategies has contributed to improving plant establishment and ensuring increased productivity in agricultural areas around the world. Here we present unprecedented results of improvement in soybean root architecture, using ammonium and phosphate fertilization.

The decrease in the rhizosphere pH with MAP was observed because its dissolution releases ammonium and phosphate ions to the soil solution, and ammonium uptake by soybean resulted in a decrease in the rhizosphere pH. Ammonium uptake by plants results in apoplast acidification in response to the activation of plasma membrane H + ATPase mediated by an AMT-dependent ammonium uptake mechanism [13], and this acidification is reflected in the rhizosphere pH. Conversely, TSP granules are composed mostly by phosphorus and calcium ions, and this explains the pH stability with this fertilizer. On average, MAP reduced the rhizosphere pH by 0.3 units (Table 1), lower than the pH decrease ranging from 0.7 to 1.4 units in maize rhizosphere observed [12] with the application of ammonium sulfate compared with calcium nitrate. We suggest that the difference in these responses was probably due to the different fertilizer rates applied in each experiment.

It has been reported that P application to soybean in P-deficient soil increases root length, dry weight, and branching [24]. The authors suggested that the main effect of P is on the lateral root initiation, and not on root elongation. In the present experiment, the application of TSP resulted in increased root length and root tips over the control (Table 2), thus corroborating results of previous reports. However, the decreased rhizosphere pH with MAP resulted in a large increase in soybean root length and in the number of root tips over TSP (Table 2). Meier and coauthors [13] showed that apoplast acidification is a driver of polarized-auxin diffusion throughout the plasma membrane, from the vascular tissue into the cortex and epidermal root cells, which favors root emergence. Moreover, even in the absence of ammonium, the authors reported that the reduction of pH solution from 5.7 to 5.0 raised auxin diffusion and lateral root emergence, showing that hydrogen is important to stimulate radial auxin movement and root proliferation. However, in the present study, ammonium uptake seemed to work as a key factor to reduce the apoplast, and eventually the rhizosphere pH, which likely led to auxin diffusion and the increase in the number of root tips [2], mainly where MAP was applied locally (Table 2). Ma and coauthors [3] showed that the localized supply of diammonium (DAP) and ammonium sulfate plus P (ASP + P) creates nutrient-rich soil patches, increasing by 50% the density and average length of lateral roots in comparison with the broadcast urea plus P, thus corroborating our findings. An ammonium-dependent effect not only increases the root proliferation but also nutrient uptake by maize [2]. They verified that the joint application of P and N reduced the rhizosphere pH and increased root and shoot growth, and that the ammonium concentration in nutrient-rich patches of the soil also stimulates N and P uptake. This is in line with our finding, since, in general, MAP$_L$ increased the P and N uptake (Figure 2) by soybean and, therefore, the PUE (Figure 3). It has been observed that the localization of P and N at the seedling and later growth stage of maize increased the N-use efficiency by 25–57% [3]. Interestingly, TSP also increased root length and the number of root tips, but the effect on P and N concentration in the plants was greater when MAP was localized, which led to the highest number of root tips. Therefore, the effect of root branching seems to be more important than just the presence of nutrient-rich soil patches. However, as root tips differed from MAP$_L$ to MAP$_B$, the fertilizer concentration seemed imperative to trigger higher root branching, thus corroborating a strict relationship between localization and root architecture, an important advance in knowledge for nutritional management strategies.

The application of both MAP and TSP superficially might have increased the interaction of the ions of ammonium and phosphate with charged soil particles, and this likely reduced the concentration of these ions near the root surface [11]. Thus, as root emergence depends on apoplast acidification triggered by ammonium uptake and consequent radial auxin movement, and the number of root tips with MAP$_B$ was lower compared with MAP$_L$

(Table 2), our results corroborate that the concentration of ammonium near the roots is very effective in boosting root emergence [11]. However, how much ammonium would be necessary remains unclear, especially for legumes, since N application could jeopardize biological nitrogen fixation [14]. Gulden and Vessey [25], assessing the influence of low ammonium concentrations as $(NH_4)_2SO_4$ on soybean nodulation 28 days after inoculation, reported that $NH_4$ exposure had a negative effect on specific nodulation (nodule number $g^{-1}$ root dry weight). However, 0.5 and 1 mM $NH_4$ resulted in a higher number of nodules per $plant^{-1}$ (NN) and N fixation, as consequence of a compensating effect on the whole plant's growth, especially roots. The authors also observed that, the higher the $NH_4$ concentration, the lower the contribution of the N derived from atmosphere to soybean. Mendes and coauthors [17] reported a decrease ranging from 15 to 50% in the number of nodes 15 days after emergence with the application of 20, 30, and 40 kg $ha^{-1}$ of N as urea, even though no effects were observed at flowering on the dry weight of nodules and soybean yields. It has been reported that the inoculation of soybean, along with P and N (90 and 25 kg $ha^{-1}$, respectively, at sowing) fertilization, increased shoot N content and P uptake [26]; this corroborates our findings, with no negative effects on the number of nodules. However, increasing the N rate to 50 kg $ha^{-1}$ impaired the inoculation efficiency and the nutrition of soybean. As such, the addition of 20 kg $ha^{-1}$ of N (120 kg $ha^{-1}$ MAP) band applied at sowing did not seem to halt N acquisition and the growth of soybean (Figures 1 and 2). The form of N in the soil solution also seems to impair BNF; however, the mechanism is not fully understood. It has been reported [27] that the lowest $N_2$ fixation occurs in soils with high nitrate contents near the root system, thus suggesting an inhibitory mechanism on BNF.

### 4.2. Phosphorus Uptake

It is known that P diffusion in soil is very slow, ranging from $10^{-12}$ to $10^{-15}$ m$^2$ s$^{-1}$ [1], and up to 90% of the P applied as fertilizer ends up adsorbed to charged mineral particles or reacting with ions such as iron, manganese, or aluminum, mainly under acidic conditions, decreasing the availability to plants. Despite the increased P uptake, P-use efficiency and soybean shoot and root growth over the non-fertilized control, applying TSP at the soil surface was not as effective as MAP or even TSP localized (Tables 2 and 3; Figures 1 and 3). For both P sources, the concentration of the fertilizer near the root system showed to be an important strategy to increase the soybean SDM once there was a significant difference from that observed where P was supplied at the soil surface (Figure 1). The soil capacity to replenish P to the depletion zone [28] near the root surface probably was hindered when P was broadcasted (Figure 2).

The increase in the PUE in the present study (Figure 3) could also be attributed, at least in part to the release of organic acids and water-soluble organic carbon, which is also stimulated by ammonium uptake [12,2 9]. These compounds of low molecular weight can supply the energy needed for microbial growth [29], thereby increasing acid phosphatase activity [30] and leading to the solubilization of less labile P forms, whereas P labile forms (microbial P) increase in the rhizosphere [12]. As a result of this chain of events, soybean plants took up more phosphorus, especially when applied close to the seed, using MAP.

Although maize is more responsive to N supply than soybean due to the extent of the biological nitrogen fixation in the latter, external supply of ammonium contributes to a better root architecture for legumes.

To our knowledge, this is the first report of ammonium decreasing the rhizosphere pH and increasing the number of root tips, shoot and root dry mass, and the PUE in soybeans. Firstly, the AMT-dependent ammonium uptake mechanism triggers the acidification of the rhizosphere, and this, in turn, generates an auxin-carrier-independent bypath across the membrane from the vasculature to the cortex and epidermal cells, facilitating the lateral root emergence (Table 2), and the expression of phosphate transporters (PHT1) at the root tips, increasing the PUE and plant growth [12,13,31–33]. Secondly, the efflux of hydrogen, as a consequence of the AMT-dependent ammonium uptake mechanism, increases the

activity of the $H+/H_2PO^-_4$ symporters dependent on the electrochemical gradient in the membrane to operate properly [2,12].

Broadcasting fertilizers is an interesting practice to optimize farm operations and explore the best planting time. However, this leads to the accumulation of inorganic and organic P close to the soil surface [34]. Our results showed that a joint application of phosphate and ammonium deeper and beside the seeds results in better root development and early plant growth. This is very important in regard to lowering the risk of crop failure due to dry spells during this first stage of plant growth, since root architecture is improved (Table 2), thus allowing the plant to better explore the soil, favoring the establishment of more resilient plants.

## 5. Conclusions

Our results showed that the joint application MAP as a source of nitrogen and phosphorus to soybean led to a decrease in rhizosphere pH, which was pivotal to change soybean root architecture and increase P acquisition and soybean carbon accrue into shoots and roots. Furthermore, regardless of the source, P applied locally is a better strategy to increase soybean PUE and root growth under tropical conditions than broadcasting the fertilizer on the soil surface. As reported in alkaline soils cropped with maize, our data also showed higher soybean root growth due to the decrease in rhizosphere pH in highly weathered acid tropical soil.

## 6. Patents

There are no patents resulting from the work reported in this manuscript.

**Author Contributions:** Conceptualization, C.A.R.; methodology, C.A.R. and J.C.C.; software, J.C.C., T.B.B. and L.V.d.M.N.; validation, C.A.R., L.V.d.M.N. and P.P.D.; formal analysis, C.A.R., J.C.C. and T.B.B.; investigation, T.B.B., P.P.D. and L.V.d.M.N.; resources, C.A.R.; data curation, C.A.R. and J.C.C.; writing—original draft preparation, P.P.D. and L.V.d.M.N.; writing—review and editing, C.A.R., J.C.C. and T.B.B.; visualization, C.A.R.; supervision, C.A.R.; project administration, J.C.C.; funding acquisition, C.A.R. All authors have read and agreed to the published version of the manuscript.

**Funding:** This work was partially supported by FAPESP, São Paulo Research Foundation, grant 2015/04200-0. CAR holds a scholarship from CNPq, National Council for Scientific and Technological Development, grant 309134/2020-0.

**Data Availability Statement:** The datasets generated during and/or analyzed during the current study are available from the corresponding author upon reasonable request.

**Conflicts of Interest:** The authors declare no conflict of interest.

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
