# Peer review of "The Joint Application of Phosphorus and Ammonium Enhances Soybean Root Growth and P Uptake"

_agriculture, doi:10.3390/agriculture12060880_

Round 1

Reviewer 1 Report

Review Comments

Based on the hypothesis that i) the application of ammonium and P jointly could change the root architecture, increase P acquisition and soybean early growth as a result of a lower rhizosphere pH; and ii) the forms of application could undermine the P use efficiency (PUE) and plant performance, this paper conducted a series of experiments to investigate the effects of joint application of ammonium and P on soybean root growth and P uptake. The study evaluates P sources applied locally or broadcasted on soybean root and shoot growth, P-use efficiency and rhizosphere pH, and unravels a win-win mechanism to the joint supply of P and ammonium to acid tropical soils. This topic is of practical significance which helps the decision-making of farmers to increase P use efficiency and soybean yields in cropped areas. However, there may be some problems with the manuscript in its present form, and some suggestions are as follows.

Point 1: Title and Abstract: the abbreviation “P in title and the first sentence in abstract is better replaced by its full name “phosphorus”.

Point 2: Abstract: The structure of the abstract part is not clear, there are no obvious distinctions among background information, study objectives and methods, results and conclusions. It’s suggested to modify this part.

Point 3: Abstract: The causality in the last sentence “Phosphorus uptake by soybean is increased early in the cycle by the joint application of P and ammonium through the decrease in rhizosphere pH resulting in root proliferation.” is not so clear, we suggest to rewrite this sentence.

Point 4: Introduction: The first and third paragraphs of the introduction part are better ending with a summarized sentence instead of stating the results of relative studies.

Point 5: Introduction: Page 1, Line 40: In this sentence “It has been shown [2,11] showed…”, the word “show” was repeated.

Point 6: Introduction: Page 2, Line 48-50: Please pay attention to the grammar of this sentence “This occurs because ammonium uptake acidifies the root apoplast, thus increasing import of protonated auxin into cortical cells and stimulating root branching [13].”.

Point 7: Results: figure 1-3: The letters showing significant difference are more appropriate with lowercase.

Point 8: Discussion: This part includes totally 9 paragraphs with a rather large extent. Thus, we suggest to add subtitles in this part.

Point 9: The framework of this paper is lack of logic, and the highlights of this study are not stressed. It is suggested to modify the whole paper including the abstract part to clarify study background and objectives, innovative findings, scientific and practical significance better.

Author Response

Point 1: Title and Abstract: the abbreviation “P” in title and the first sentence in abstract is better replaced by its full name “phosphorus”.

Edited

Point 2: Abstract: The structure of the abstract part is not clear, there are no obvious distinctions among background information, study objectives, and methods, results and conclusions. It’s suggested to modify this part.

The abstract was thoroughly edited

Point 3: Abstract: The causality in the last sentence “Phosphorus uptake by soybean is increased early in the cycle by the joint application of P and ammonium through the decrease in rhizosphere pH resulting in root proliferation.” is not so clear, we suggest to rewrite this sentence.

The sentence was rewriten

Point 4: Introduction: The first and third paragraphs of the introduction part are better ending with a summarized sentence instead of stating the results of relative studies.

Edited accordingly

Point 5: Introduction: Page 1, Line 40: In this sentence “It has been shown [2,11] showed…”, the word “show” was repeated.

Corrected

Point 6: Introduction: Page 2, Line 48-50: Please pay attention to the grammar of this sentence “This occurs because ammonium uptake acidifies the root apoplast, thus increasing import of protonated auxin into cortical cells and stimulating root branching [13].”.

The sentence was rewriten

Point 7: Results: figure 1-3: The letters showing significant difference are more appropriate with lowercase.

In figure 1 Capitals and lower case compare different results. In figures 2 and 3 it was changed.

Point 8: Discussion: This part includes totally 9 paragraphs with a rather large extent. Thus, we suggest to add subtitles in this part.

We added subtitles

Point 9: The framework of this paper is lack of logic, and the highlights of this study are not stressed. It is suggested to modify the whole paper including the abstract part to clarify study background and objectives, innovative findings, and scientific and practical significance better.

The innovative findings and practical relevance are highlighted in the two last paragraphs of the manuscript.

Reviewer 2 Report

This study is about enhancing of soybean root growth and phosphorus uptake by joint application of phosphorus and ammonium. The results clearly show that joint application of phosphorus and ammonium is more beneficial for soybean root and shoot growth comparing with other fertilizers. Moreover, authors compared two types of application of a fertilizer: localized and broadcasted, and revealed that the first one results in higher phosphorus use efficiency. In my opinion, I have some comments.

Line 2 – it should be “phosphorus” instead of P, because it is the first mention.

Line 7 –it should be on the line 9.

I hope editors will help you with correct filling the form

Line 27 – near the word “Phosphorus” abbreviation (P) is needed, because later you use P and it is not clear what you mean. In my opinion, Al and Fe do not need the abbreviation. You use only once.

Line 38 – it is better to say “Gamuyao with authors”. The same is for lines 54, 208, 216, 249

Line 40 – something is wrong with ‘shown’ and ‘showed’ and the reference is on the wrong place.

Line 84 – Did you use only two seeds of soybean? At line 84 – two seeds, later at line 86 – 50kg of seeds. So how many seeds did you use? What is 100 ml 50 kg-1? 100 ml of bacteria suspension per 50 kg of seeds? Later at line 87, you give the concentration of Rhizobia per gram. Per gram of what? Is it a concentration of bacteria cells in inoculation suspension? In this case it should be “per 1 ml”

According to other articles (below) strains SEMIA 587 and SEMIA 5019 belong to B. elkanii.

Melissa, O., Cristian, A., Sheila, C. et al. Evaluation of nitrous oxide emission by soybean inoculated with Bradyrhizobium strains commonly used as inoculants in South America. Plant Soil 472, 311–328 (2022). https://doi.org/10.1007/s11104-021-05242-y

Zilli, J.É., Pacheco, R.S., Gianluppi, V. et al. Biological N2 fixation and yield performance of soybean inoculated with BradyrhizobiumNutr Cycl Agroecosyst 119, 323–336 (2021). https://doi.org/10.1007/s10705-021-10128-7

Cerezini, P., Kuwano, B.H., Grunvald, A.K. et al. Soybean tolerance to drought depends on the associated Bradyrhizobium strain. Braz J Microbiol 51, 1977–1986 (2020). https://doi.org/10.1007/s42770-020-00375-1

Are you sure that you used B. japonicum? In addition, it should be mentioned that these strains are commercial. Or give an gene bank or bacteria collection where did you get these strains.

Is it possible to give the level of humidity in a greenhouse during plant growth?

Line 159 – at figure 1, MAPL differs from MAPB but there is no significant difference between TSPL and TSPB. That is why it is better not to say that root dry matter yields higher if fertilizer is localized.

Author Response

Line 2 – it should be “phosphorus” instead of P, because it is the first mention.

Corrected

Line 7 –it should be on the line 9.

I think now it is correct

I hope editors will help you with correct filling the form

Line 27 – near the word “Phosphorus” abbreviation (P) is needed, because later you use P and it is not clear what you mean. In my opinion, Al and Fe do not need the abbreviation. You use only once.

OK

Line 38 – it is better to say “Gamuyao with authors”. The same is for lines 54, 208, 216, 249

Edited

Line 40 – something is wrong with ‘shown’ and ‘showed’ and the reference is on the wrong place.

Corrected

Line 84 – Did you use only two seeds of soybean? At line 84 – two seeds, later at line 86 – 50kg of seeds. So how many seeds did you use? What is 100 ml 50 kg-1? 100 ml of bacteria suspension per 50 kg of seeds? Later at line 87, you give the concentration of Rhizobia per gram. Per gram of what? Is it a concentration of bacteria cells in inoculation suspension? In this case it should be “per 1 ml”

The text was edited for clarity

According to other articles (below) strains SEMIA 587 and SEMIA 5019 belong to B. elkanii.

Melissa, O., Cristian, A., Sheila, C. et al. Evaluation of nitrous oxide emission by soybean inoculated with Bradyrhizobium strains commonly used as inoculants in South America. Plant Soil 472, 311–328 (2022). https://doi.org/10.1007/s11104-021-05242-y

Zilli, J.É., Pacheco, R.S., Gianluppi, V. et al. Biological N2 fixation and yield performance of soybean inoculated with BradyrhizobiumNutr Cycl Agroecosyst 119, 323–336 (2021). https://doi.org/10.1007/s10705-021-10128-7

Cerezini, P., Kuwano, B.H., Grunvald, A.K. et al. Soybean tolerance to drought depends on the associated Bradyrhizobium strain. Braz J Microbiol 51, 1977–1986 (2020). https://doi.org/10.1007/s42770-020-00375-1

Are you sure that you used B. japonicum? In addition, it should be mentioned that these strains are commercial. Or give an gene bank or bacteria collection where did you get these strains.

Sorry, this was a mistake. Thank you very much for pointing it out. It was corrected

Is it possible to give the level of humidity in a greenhouse during plant growth?

We do not have this.

Line 159 – at figure 1, MAPL differs from MAPB but there is no significant difference between TSPL and TSPB. That is why it is better not to say that root dry matter yields higher if fertilizer is localized.

Thank you for pointing it out. Corrected.

Round 2

Reviewer 1 Report

The authors have addresses the suggestions in the revised version